# HexFire: A Flexible and Accessible Wildfire Simulator

**Nathan H. Schumaker** [1,*] , **Sydney M. Watkins** [2] **and Julie A. Heinrichs** [3]

1    U.S. Environmental Protection Agency, Center for Public Health and Environmental Assessment, Corvallis, OR 97333, USA

2    Oak Ridge Institute for Science and Education, % U.S. EPA Center for Public Health and Environmental Assessment, Corvallis, OR 97333, USA

3    Computational Ecology Group Inc., Canmore, AB T1W 1P2, Canada

*    Correspondence: schumaker.nathan@epa.gov

**Abstract:** As fire frequency and severity grow throughout the world, scientists working across a range of disciplines will increasingly need to incorporate wildfire models into their research. However, fire simulators tend to be highly complex, time-consuming to learn, and difficult to parameterize. As a result, embracing these models can prove impractical for scientists and practitioners who are not fire specialists. Here we introduce a parsimonious wildfire simulator named HexFire that has been designed for rapid uptake by investigators who do not specialize in the mechanics of fire spread. HexFire should be useful to such nonspecialists for representing the spread of fire, interactions with fuel breaks, and for integrating wildfire into other types of ecological models. We provide a detailed description of the HexFire simulator's design and mechanisms. Our heuristic fire spread examples highlight the flexibility inherent in the model system, demonstrate that HexFire can generate a wide range of emergent fire behaviors, and illustrate how HexFire might be coupled with other environmental models. We also describe ways that HexFire itself might be altered or augmented. HexFire can be used as a proxy for more detailed fire simulators and to assess the implications of wildfire for local ecological systems. HexFire can also simulate fire interactions with fuel breaks and active fire suppression.

**Keywords:** fire model; fire simulation; fire management; fire impacts; HexSim; HexFire

## 1. Introduction

Many complex and interacting mechanisms shape the rates and patterns of wildfire spread [1–6]. Decades of investigation have revealed much about these processes, and the insights derived from this work have led to the development of increasingly realistic wildfire models [7–15]; but despite their advantages, these detailed models will not always constitute the best solution to a problem as they tend to be hard to learn, difficult to parameterize, and tedious to repurpose. Although many scientific and management challenges are best addressed using the most advanced analytic tools and methods, the benefits of parsimony can at times prove compelling. Simpler solutions may be more appropriate if data are limiting, or when a less detailed proxy for a complex process is desirable or necessary.

Many rigorous wildfire simulation models have already been developed and made available. Examples include FIRETEC [9], FARSITE [16], FlamMap [17], Prometheus [18], FSim [19], CanFire [20], BorealFireSim [21], QUIC-fire [22], and ABWiSE [23]. However, from a practical standpoint these and other complex applications cannot be readily adopted by researchers with cursory experience in wildfire mechanics, whose interests and data are primarily focused on other disciplines, or who have limited time to invest in learning new software. Here we introduce a wildfire simulation model, called *HexFire*, that is free, open-source, low-parameter, easy to learn, and readily adaptable to new locations or systems. We expect HexFire will contribute to ecological and conservation-related studies

that require a wildfire model, but for which the forecasting of highly specific fire spread and behavior is not the primary focus of model design. HexFire also facilitates the exploration of competing wildfire suppression strategies.

HexFire makes use of cellular automata and individual-based modeling techniques to simulate fire spread, both locally and across long distances. The model requirements include eight parameters that affect simulated burn rates and ember transport, and input maps describing the known or presumed flammability of available fuels, wind speed and direction, plus a few other system attributes. Below, we describe the HexFire model design and mechanisms, and provide three examples that illustrate its behavior, strengths, weaknesses, and applications. The HexFire model, data sets, and utilities used in this study can all be downloaded from www.hexsim.net (accessed on 10 July 2022). We have also developed a downloadable video demonstration illustrating the mechanics of running and modifying HexFire (see Supplemental Materials).

## 2. Materials and Methods

We constructed HexFire within the HexSim model development platform. HexSim [24] is a well-established software application that has been used in >60 peer-reviewed publications. Although HexSim models have traditionally been designed to conduct population viability analysis for species of conservation concern [25–34], use of the platform has expanded to a wider range of topics (e.g., disease models [35]). HexSim and HexFire are spatially explicit and can easily incorporate multiple static or dynamic maps quantifying the amounts and patterns of fuels, moisture levels, wind, ignition sites, management interventions, and other drivers relevant to wildfire.

### 2.1. Model Overview

HexFire is an individual-based model (IBM) that functions in part as a cellular automaton [36]. Cellular automata (CA) are grid-based models in which all cells are assigned one of a finite number of states. Each time step, every cell's state is updated based on its present state, and that of a pre-defined collection of neighbors. Interest in these simple models originally stemmed from their ability to generate complex spatial dynamics [37]. While CA models are typically implemented on square grids, using hexagonal tessellations (as is done here) can enhance their realism because then all immediate neighbors become equidistant. To replicate CA model functionality, we employ automata (hereafter, *automata*, *agents*, or *individuals*) to record and update each cell's state (unburned, burning, burned). In addition, HexFire uses HexSim individuals to simulate the long-distance movement of embers. As a result, HexFire's design has elements in common with state and transition simulation models (STSMs, [38]); and though our model's simplicity results from its being more like a CA than a STSM, HexFire is actually a spatial IBM. It does not have the limitations inherent with STSMs, such as the inability to track continuous-valued state variables, assign complex behaviors to independent agents, or to impose covariance structures between distinct sets of transition probabilities [38].

We designed HexFire to be easy to parameterize and run. The data required by the model include eight parameter values (Table 1) and three principal input maps (Table 2). These maps describe the distribution of fuel flammability, wind speed, and wind direction. Our first example illustrates HexFire's behavior when used with simplistic versions of these three maps; but we employ more realistic-looking maps in our second and third examples. HexFire also requires five ancillary input maps (described below), but their construction involves minimal effort.

**Table 1.** HexFire input parameters.

| Parameter Name | Parameter Interpretation |
|---|---|
| Burn Iterations per Time Step | The number of times that the contagious and ember-driven fire spread algorithms are run per time step. |
| Iterates to Burn Completely | The number of burn iterations during which an ignited cell will continue to burn. |
| Flammability Exponent | The exponent that influences how fuel flammability affects contagious wildfire spread. |
| Proximity Exponent | The exponent that influences how the number of burning neighbors affects contagious wildfire spread. |
| Ember Creation Rate | The maximum number of embers that can be created, per iterate, within each burning cell. |
| Ember Max Distance | The maximum distance, in hexagons, that an individual ember may travel. |
| Ember Step Length—Wind | The step length, in hexagons, assigned to embers moving along a wind gradient. |
| Ember Step Length—Random | The step length, in hexagons, assigned to embers moving in a random direction. |

**Table 2.** Input maps used by the HexFire model. *Variable* suggests that map construction effort will range from minimal to significant, depending on study design. *Automatic* implies that a method or utility is available to construct the map. *Optional* indicates that a map may be left blank.

| Map Name | Level of Effort | Map Function |
|---|---|---|
| Relative Flammability | Variable | Provides the relative flammability of each cell in the landscape. Values must range between 0 and 1. |
| Ignition Sites | Variable | Controls the time and location at which fires are initiated, including back burns. |
| Hexagon ID | Automatic | Contains the individual ID of each cell. This map is trivial to create in HexSim. |
| Patch Maps (A-D) | Automatic | A collection of four patch maps for which the union of all patches is space-filling, and each patch slightly overlaps its neighbors. We provide a utility for constructing these patch maps, which are used to improve model performance. |
| Relative Wind Speed | Variable | Provides the wind speed for each cell in the landscape. Values must range between 0 and 1. |
| Wind Gradient | Variable | Indicates the directions that embers will travel. We provide a utility that builds wind gradient maps from maps of wind direction. |
| Fuel Breaks | Optional | Indicates where and when fuels should be removed from the flammability map. |
| Fuel Barriers | Optional | Specifies the location of fuel barriers, which can block the movement of embers. |

Though HexFire's input flammability maps may contain only data on fuels, users may instead supply a rate of spread (ROS) map that additionally captures moisture content, wind speed, wind direction, atmospheric conditions, and/or topography. When a ROS map is not being used, information describing wind speed and direction may still be supplied via separate input maps. However, HexFire was not designed to directly incorporate maps of topography or atmospheric conditions—the direct use of these types of data would necessitate making some model modifications. In addition to providing input parameters and maps, customizing HexFire for a new system will require selecting a hexagon size, specifying the study area dimensions, and constructing a HexSim workspace (see Supplemental Materials).

Simulated wildfires begin spreading after one or more fires are initiated in locations indicated by a temporally dynamic map of ignition sites. Fires subsequently spread via contagion and ember transport, and fuel combustion status is represented internally within each hexagon (hereafter, *hexagon*, *cell*, or *location)*. HexFire has an implicitly defined time step and a user-defined simulation duration. Here, we assume that a time step corresponds to one hour but emphasize that this interpretation will vary between systems and studies. To increase model efficiency, HexFire agents are automatically added to the landscape on an as-needed basis, while a simulation runs.

HexFire simulates wildfire spread using two principal mechanisms: the expansion of existing fires within their immediate neighborhood (contagious spread), and the movement of embers, which can subsequently initiate new fires at distant locations (fire spotting). Multiple iterations of contagious and ember-driven spread may be conducted each time step, with this frequency being specified via a *Burn Iterations per Time Step* parameter. This parameter allows users to control the rate of fire spread, which in actual wildfires would vary based on the type of fuels present, moisture content, wind, and other factors. Once a cell catches fire, it continues to burn for a period specified by an *Iterates to Burn Completely* parameter. In actual fires, this time period would be affected largely by the quality and moisture content of available fuels. In HexFire, this parameter allows users to adjust the amount of time during which a burning cell contributes to new fires via contagious or ember-driven spread. The *Iterates to Burn Completely* parameter may be smaller, larger, or equal to *Burn Iterations per Time Step*.

The rate of contagious fire spread is governed by the equation

$$P_{contagion} = P_{proximity} \times P_{flammability}, \tag{1}$$

where $P_{proximity}$ and $P_{flammability}$ are hexagon-specific values derived in part from HexFire model state (proximity) and input maps (flammability). For every unburned cell that has at least one neighbor presently on fire, we compute the value of $P_{contagion}$, draw a uniformly distributed random number R between 0 and 1, and initiate a new fire if $P_{contagion} > R$. To accomplish this, we define

$$P_{proximity} = 1 - (1 - N/6)^{\alpha}, \tag{2}$$

where N refers to the number of a cell's immediate neighbors that are currently burning (a value between 0 and 6), and $\alpha$ is our *Proximity Exponent* parameter. We also define

$$P_{flammability} = 1 - (1 - F)^{\beta}, \tag{3}$$

where F refers to an unburned cell's value in the Relative Flammability input map, and $\beta$ is our *Flammability Exponent* parameter. The Relative Flammability map is intended to capture the availability of fuels on the landscape, and it must contain values that range between 0 (no fuel) and 1 (the most flammable fuel). The Relative Flammability map should be static because its values are assigned to each automaton once and are never updated. As long as $\alpha$ and $\beta$ exceed zero, the functions for $P_{proximity}$ and $P_{flammability}$ will generate families of nontrivial probability curves (Figure 1).

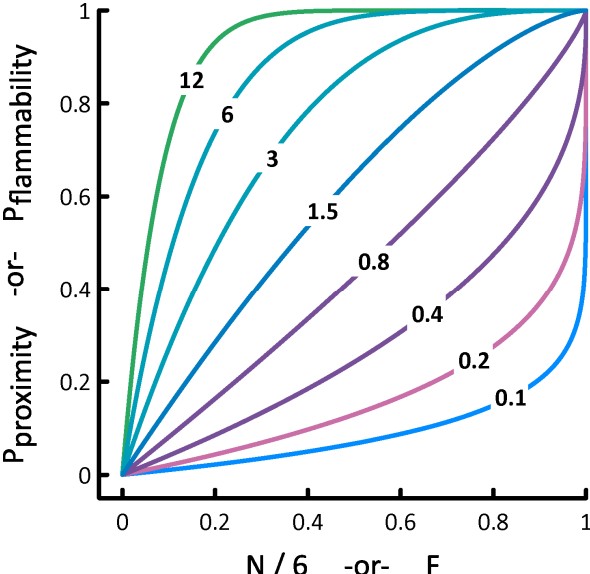

**Figure 1.** Shapes of the probability curves derived from specific values of N (neighbors on fire) or F (relative flammability), and α or β (the proximity and flammability exponents). Exponent values are superimposed on the curves they produce.

Because $P_{contagion}$ is the product of $P_{proximity}$ and $P_{flammability}$, HexFire's local fire spread dynamics can vary widely (Figure 2). Through judicious choice of the proximity and flammability exponents, users can make contagious fire spread more or less sensitive to a cell's relative flammability, or its number of burning neighbors. Alternatively, users may simply replace our functions governing contagious fire spread with their own (see Supplemental Materials).

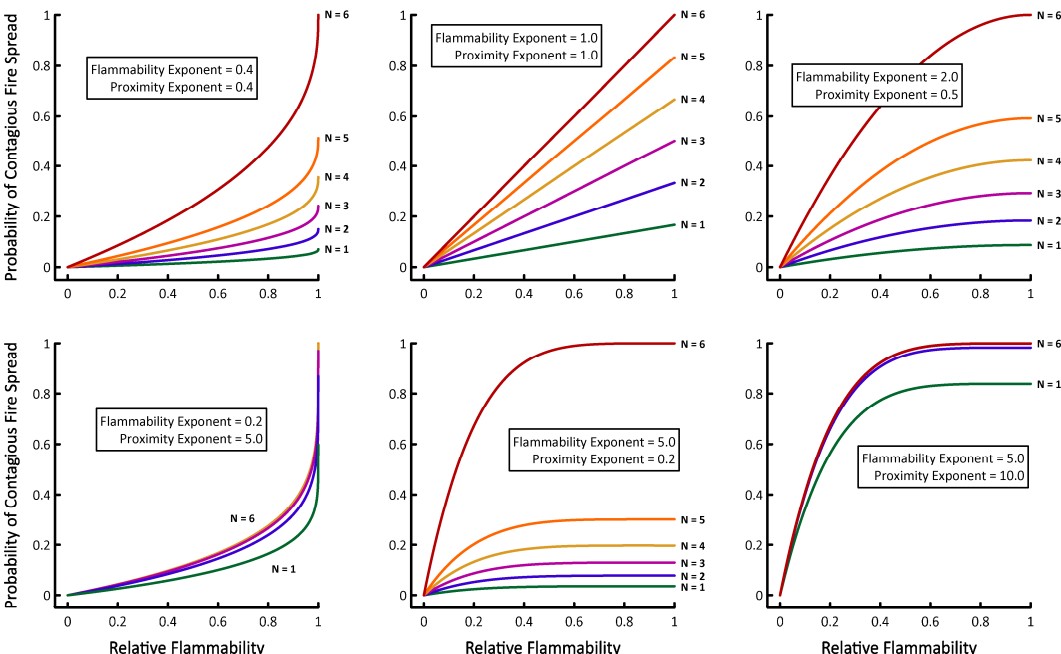

**Figure 2.** Six examples of how the probability of contagious fire spread changes as a function of a cell's relative flammability, and its number of burning neighbors (N).

Emergent rates of fire spotting are governed by parameters that control the production and movement of embers. At each cell that is actively burning, we add E embers, where

$$E = F \times \varepsilon. \tag{4}$$

Here, $\varepsilon$ is a parameter we refer to as the *Ember Creation Rate*, and F (discussed above) represents a cell's relative flammability. The *Ember Creation Rate* sets the maximum number of embers that can be created per iterate, for each burning cell. The number of embers actually created in a cell (within a single iterate) will range between zero and this maximum value. E is always rounded to an integer. Each ember will then move a distance D (in hexagons) defined by

$$D = S \times \delta, \tag{5}$$

where S is the location-specific relative wind speed (see below), and $\delta$, our *Ember Max Distance* parameter, sets the maximum distance in hexagons that embers are allowed to travel before settling into a distant cell. The distance that any given ember actually travels will range between zero and this maximum value. D is always rounded to an integer. As embers move, they take successive steps along the wind gradient, and then in a random direction, until they reach their assigned movement distance. The lengths of these gradient-following and random steps are controlled by the *Ember Step Length—Wind* and *Ember Step Length—Random* parameters. The addition of random steps keeps the ember's movements from becoming overly deterministic. When embers land, they initiate new fires at a rate specified by $P_{flammability}$, as defined above. The mechanisms that HexFire uses to simulate fire spotting may be easily modified (see Supplemental Materials).

HexFire incorporates information on wind speed and direction via two input maps. The Relative Wind Speed map should contain values between 0 (no wind) and 1 (maximum wind velocity) and is used in computing the distance that embers will travel. Similarly, the Wind Gradient map must depict gradients that specify which directions embers should travel in; embers will move from low to high gradient values. Both the Relative Wind Speed and Wind Gradient maps may be dynamic. We have developed an easy-to-use C-language utility that constructs wind gradient input maps from maps of wind direction (see Supplemental Materials).

Three additional input maps specify wildfire ignition points, plus the timing and location of fuel breaks and fuel barriers. The Ignition Sites map can be used to add fire to the landscape when a simulation begins, and at any subsequent time during a simulation, including when back burns are initiated. The Fuel Breaks and Fuel Barriers input maps must be included in HexFire simulations, but they may be left blank. When utilized, these maps make it easy to simulate static or spatially and temporally dynamic fire suppression activities. Nonzero cells in the Fuel Breaks map indicate where and when fuels will be automatically removed from the landscape. Fuel Barrier maps describe linear features that can repel embers; they are intended to be used in conjunction with back burns to simulate active management that effectively limits the directions in which these intentionally set fires may spread. Thus, we expect that Fuel Breaks be used to simulate the manual removal of fuels, where the exact extent of the clearing is known. We expect Fuel Barriers to be used to control the directions in which simulated back burns spread, allowing users to impose some constraints upon a process whose outcomes cannot be precisely known.

HexFire also makes use of two additional input maps to deal with ancillary tasks. First, the model requires a map containing the ID of each hexagon. This type of map is trivial to create in HexSim. Second, HexFire relies on a set of four patch maps for which: (a) the union of all patches is space-filling; and (b) each patch slightly overlaps its neighbors. HexSim model performance scales with population size, and these maps improve simulation speeds by allowing HexFire to add new agents only when they are needed to simulate fire. We developed an easy-to-use C-language utility that constructs these sets of overlapping patch maps (see Supplemental Materials).

Finally, HexFire has optional settings that may be used to initiate fires at randomly selected locations when a simulation begins, to make the model terminate if no active fires are detected, and to control the types of output maps the model generates (see Supplemental Materials).

We developed three heuristic examples that illustrate applications of HexFire. Our example study areas are all 250 K hexagonal cells in extent (500 rows × 500 columns), and because they are fabricated, we avoid using absolute spatial scales (providing dimensions in kilometers or specifying hectares per hexagon). We kept the spatial extent of the examples small in order to simplify the illustrations, and we anticipate that real-world applications of HexFire will make use of much larger maps.

### 2.2. Example 1—Model Parameters

Our first example explores how observed rates and patterns of fire spread respond to changes in the parameters that control contagion and ember transport. We began with a baseline set of parameters (Table 3) and made use of highly simplified input maps depicting relative flammability, relative wind speed, and a wind gradient. In this example, the relative flammability map is made up from a 50 × 50 array of square blocks, each of which is 10 × 10 hexagons in size. All hexagons within a single block are assigned the same value, and block scores alternate between 1.00 (high flammability) and 0.25 (low flammability), in a checkerboard pattern. The juxtaposition of low and high flammability blocks is intended to illustrate the impacts of fuel flammability on contagious wildfire spread. The relative wind speed map was set to 1 in every hexagon, and the wind gradient always directed embers due south, towards the lower edge of the simulation landscape. We initiated the simulated fires at the landscape's top-center, and did not make use of fuel breaks, fuel barriers, or back burns.

**Table 3.** Example 1 input parameters. Labels used to distinguish the six experiments are shown in parentheses. Experiments B–F were conducted using the baseline model (experiment A) parameters, except where indicated.

| | Baseline | Model Variants | | | | |
|---|---|---|---|---|---|---|
| | (A) | (B) | (C) | (D) | (E) | (F) |
| Burn Iterations per Time Step | 2 | | | | | |
| Iterates to Burn Completely | 3 | | | | | |
| Flammability Exponent | 2 | 5 | 0.2 | | | |
| Proximity Exponent | 0.5 | 0.2 | 5 | | | |
| Ember Creation Rate | 5 | | | | | |
| Ember Max Distance | 10 | | | 50 | | |
| Ember Step Length—Wind | 1 | | | | 10 | |
| Ember Step Length—Random | 1 | | | | | 10 |

For this initial example, we ran 100 replicates each of six HexFire model variants (Table 3). Every replicate was run for one day (24 time steps). We developed a baseline model (A) and five alternate models (B–F) that varied one or more input parameters. Model variants B and C examined the influence on fire spread of alterations to the flammability and proximity exponents (Figure 2). Variants D, E, and F modified the rules governing ember movement.

### 2.3. Example 2—Fire Suppression

We developed a second example to illustrate how HexFire may be used to simulate wildfire responses to fuel breaks. This example was run with our baseline parameter values (Table 3), but utilized more realistic input maps depicting relative flammability, relative wind speed, and wind gradient (Figure 3). To illustrate fire suppression, we installed two fuel breaks in locations that were informed by simulations without fuel breaks. We ran 100 replicate simulations with, and without fuel breaks. Every replicate was run for one week (168 time steps).

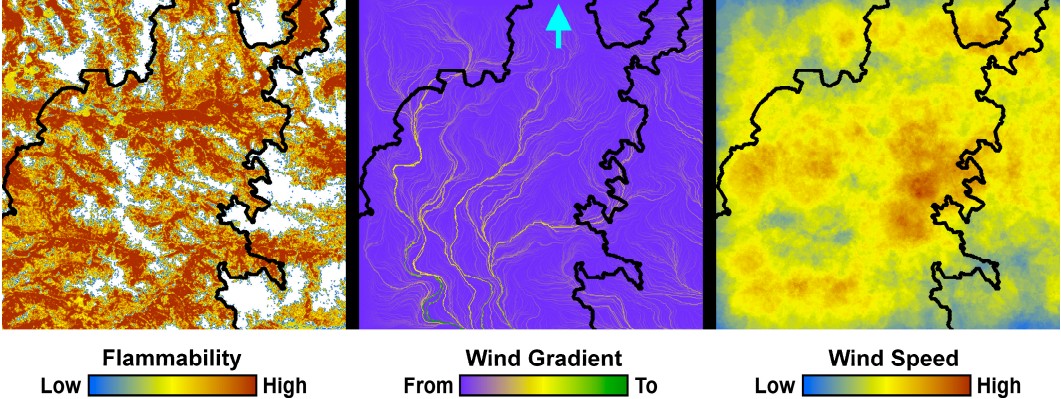

**Figure 3.** Example 2 spatial inputs, including relative flammability (**left**), wind gradient (**center**) and relative wind speed (**right**). The emergent simulated burn perimeter (see Results) is shown as a black outline. The arrow at the top-center indicates the location at which the fire was initiated.

This experiment's map of relative flammability was derived from an actual image of a forested landscape. However, the method we employed for converting forest cover classes into flammability estimates was intentionally arbitrary. Our goal was simply to create a spatial distribution of flammability values that was plausible in appearance, relative to the one used in our first example. Similarly, the wind gradient and wind speed maps we developed for this example are complex and realistic in appearance but were not derived from empirical wind data. This example did not utilize fuel barriers or back burns, and the map used to initiate fires was the same as that of example 1.

### 2.4. Example 3—Coupled Models

Our intent in this example was to explore the use of HexFire as a disturbance generator that influences another ecological process. To this end, we integrated HexFire into an existing pine marten (*Martes martes*) IBM. We implemented this pine marten model within the same 500 × 500 hexagon HexSim workspace used in the previous examples. However, the marten model utilized just under half the overall landscape's width; specifically, it occupied 119 K of 250 K hexagons total. Of these, 33,667 hexagons (28%) contained marten habitat (Figure 4). To facilitate this merging of HexSim workspaces, we fabricated a novel map of marten habitat quality. We therefore refer to this simulator below as simply a generic marten model.

Our marten simulator is a two-sex, three stage class model that includes sex-specific defended areas, or territories. Within a sex, territories may not overlap; but overlap is allowed between sexes. Marten territories vary in size between 8 and 120 hexagons, depending on resource quality and availability, and the simulated martens acquire resources from these defended areas. Stage class (juvenile, subadult, adult) and resource allocation together determine the emergent survival and reproductive rates. The marten model simulates mate-finding, and only paired females may reproduce. Subadult martens disperse from their natal site prior to territory acquisition. The model's time step is one year.

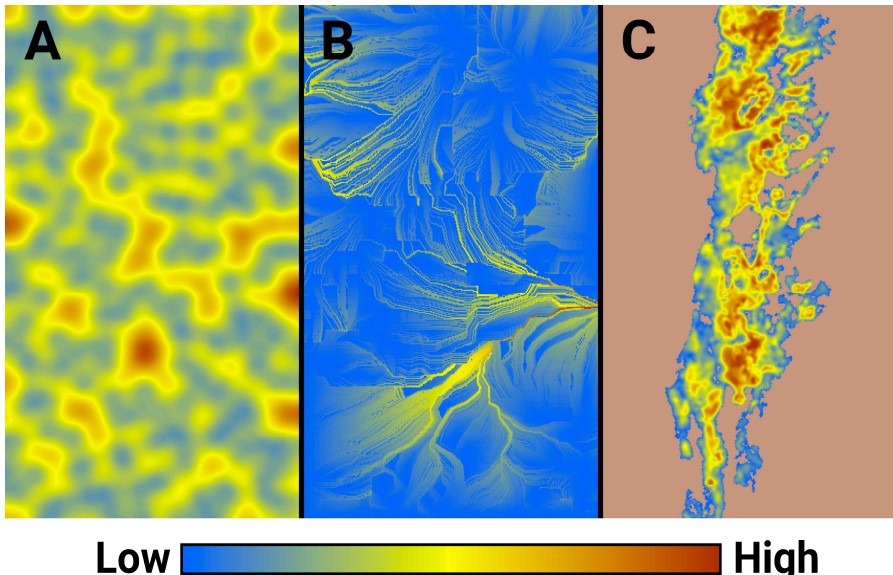

**Figure 4.** Example 3 spatial inputs, including (**A**) relative flammability, (**B**) wind gradient, and (**C**) marten habitat quality.

We initially ran 100 replicates of the marten model, each with a randomly distributed starting population. We let all of these simulations proceed to steady-state, and then periodically gathered snapshots of the resulting model state. From this large collection, we selected a single snapshot that reflected a stable population slightly above the emergent carrying capacity and used that data set to initialize all subsequent marten simulations. This process, which is straightforward to implement in HexSim, was intended to remove startup bias from subsequent comparisons of model treatments.

Although the coupling of HexFire and HexSim models can be seamless, we indirectly linked the fire and marten simulators to demonstrate how users may simplify the integration of models having different temporal increments—here, hourly vs. yearly time steps. To join the models, we ran many HexFire simulations, assembled a large collection of burn outcomes, and imported the results into the marten model. We began each HexFire simulation using a single randomly located ignition site, and let the fires burn until they went out. We then collected maps showing the ending burn configuration resulting from each of these trials. In a minority of cases, the simulated fires would fail to catch when the model started up. To address these trivial outcomes, we removed any results corresponding to a final burn area of less than 100 hexagons. These were replaced with results from additional replicates, until the total number of nontrivial burn outcomes was 1000.

We ran the fire simulations using the baseline parameter values (Table 3) with two modifications. We set *Burn Iterations per Time Step* to 1 (from 2) and *Iterates to Burn Completely* to 5 (from 3). These changes made the simulated fires burn slowly but consistently. We then modified some of the optional model settings, turning the minimum and maximum number of random fires to 1, setting the early termination switch on, and instructing the model to generate only the final burn map. For this example, we supplied HexFire with a fabricated map of relative flammability (Figure 4). We constructed this map by placing random points onto an otherwise blank image, iteratively smoothing the result, and then transforming it using a nonlinear function that exaggerated the number of large values. We developed a wind gradient map (Figure 4) from a small bit of a simulated wind direction image that had been assembled for other purposes. We also set all hexagons in the ignition sites input map to zero. Otherwise, the fire-related input maps were the same as those employed in example 1.

We ran the marten life history simulator with and without wildfire. When wildfire was included, we always began the marten simulations by selecting a single burn configuration map from the collection of 1000 realizations described above. We set the marten

habitat quality values to zero throughout the burned area, and removed any martens located there. This simulated a worst-case scenario within which martens were unable to escape a wildfire that completely decimated any habitat resources in their path. We ran 100 replicate simulations of the model without wildfire, and 1000 replicates of the model that incorporated wildfire disturbance. In all cases, the marten simulations were run for a period of 50 years. Example 3 did not make use of fuel breaks, fuel barriers, or back burns.

## 3. Results

### 3.1. Example 1—Model Parameters

We used our initial example to explore the consequences of altering select HexFire input parameters. Our six model variants generated estimates of burn frequency that exhibited similarities and differences (Figure 5). Mean burn frequency was always computed on a per-hexagon basis, as the number of replicates in which a hexagon burned divided by the total number of replicates. We ran 100 replicates of each model variant and examined each result after the simulated fires had burned for one day (24 time steps). We have developed an easy-to-use C-language utility that generates maps of mean burn frequency from HexFire model output (see Supplemental Materials).

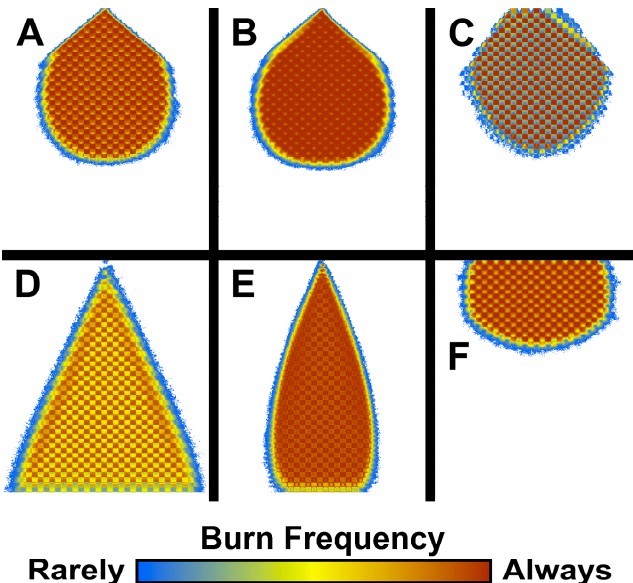

**Figure 5.** Mean burn frequencies produced by 100 replicates of six model variants: (**A**) the baseline model, (**B**) increased influence of proximity to fire, (**C**) increased influence of fuel flammability, (**D**) longer ember distances, (**E**) enhanced wind influence, (**F**) diminished wind influence. The checkerboard patterns derive from the model's flammability map, which was made up from alternating blocks of low and high flammability fuels.

Relative to the baseline parameters, the overall mean burn frequency increased when the *Flammability Exponent* was increased and the *Proximity Exponent* decreased (experiment B). The mean burn frequency decreased when these parameter modifications were inverted (experiment C). Not surprisingly, the mean fire footprint expanded significantly when embers were allowed to travel further (experiment D). Likewise, the fire footprint narrowed when embers were made to more closely track the wind direction (experiment E) and widened when embers traveled more randomly (experiment F).

To test the insights derived from these results, we conducted an additional experiment that incorporated the proximity and flammability exponents used in experiment B, the ember movement distance used in experiment D, and the ember step length parameters used in experiment F. We ran 100 replicates of this hybrid model, and (as anticipated) observed that the mean fire footprint expanded to almost the entire simulation landscape.

### 3.2. Example 2—Fire Suppression

Our second example was designed to illustrate HexFire's ability to simulate fire suppression. We began by running 100 replicates of this experiment without fuel breaks. In the absence of fire suppression, 44% of the burnable area remained unburnt in every simulation (Figure 6). Just under 9% of the burnable area was consistently consumed by fire (burned in at least 90 of 100 replicates).

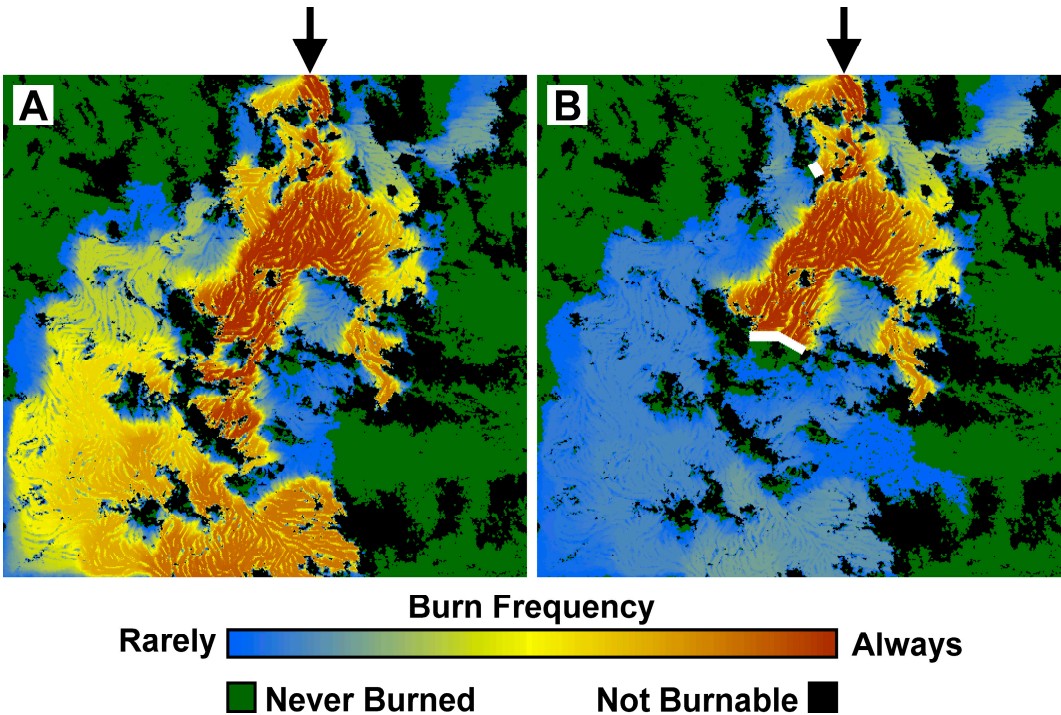

**Figure 6.** Mean burn frequencies (**A**) without and (**B**) including fuel breaks. Fuel breaks, shown in white, indicate areas where the fuel flammability was set to zero. Black arrows show the location at which the fires were initiated. These results were assembled from 100 replicate simulations.

By inspecting the dynamic emergent spatial patterns of fire progression produced by these initial simulations, we identified two locations where it seemed likely that fuel breaks would significantly limit the overall area consumed by fire. We placed 10-hexagon wide fuel breaks in these two locations (Figure 6) and ran 100 additional replicates of this model variant. Our *Ember Max Distance* parameter prevented embers from travelling directly across these fuel breaks.

In spite of their being generally effective, our two fuel breaks did sometimes fail to arrest the fire's spread. In those cases, small but persistent fires managed to burn around the fuel breaks, and eventually merged to form large burns. This result illustrated the variability that emerges naturally from the heterogeneity in our input maps (Figure 3), coupled with the stochasticity inherent in HexFire's fire spread mechanisms.

### 3.3. Example 3—Coupled Models

As described in the Methods, we used HexFire to generate a collection of 1000 fire outcomes, each corresponding to a single simulated wildfire season. The resultant distribution of observed fire sizes was dominated by small burns but exhibited a large number of intermediate-sized disturbances as well (Figure 7). The largest of our simulated burns consumed just under 47% of the entire landscape. These wildfires were unequally distributed across the study landscape, and the most frequently burned areas fell outside the marten's range (Figure 8).

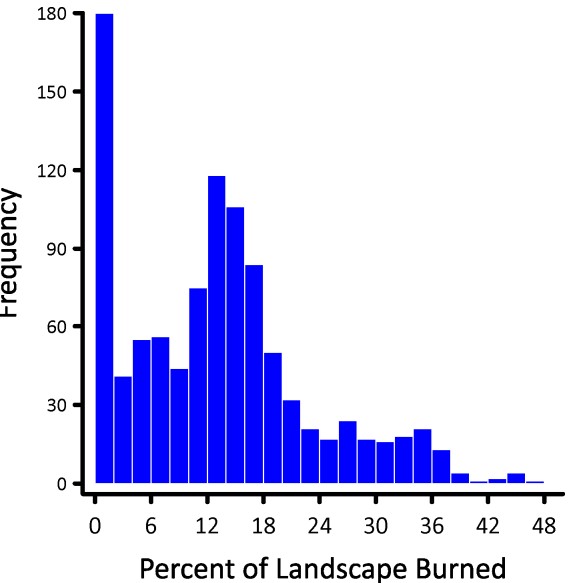

**Figure 7.** The emergent distribution of burn areas observed from 1000 fire simulations. Replicates that generated a fire size of less than 100 hexagons (of 119 K hexagons total) were replaced with the results from an additional simulation.

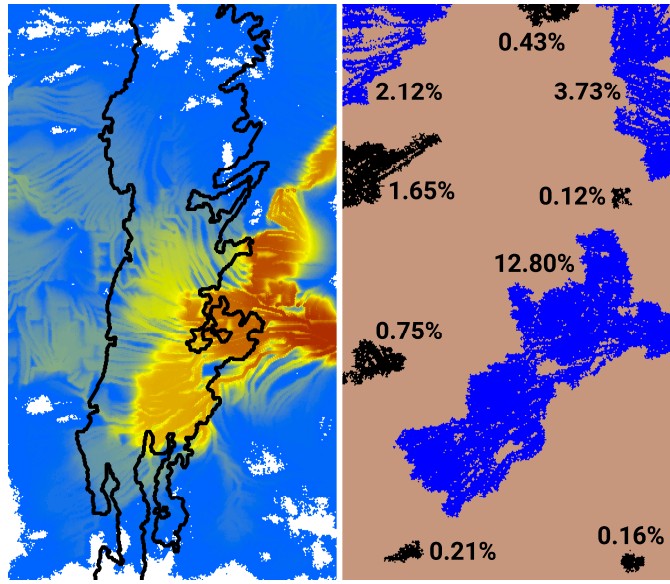

**Figure 8.** Mean burn frequency generated from 1000 fire simulations (**left**), and nine illustrative non-overlapping fires selected from this collection (**right**). Mean burn frequency is displayed using the color ramp of Figure 4, and the black outline indicates the extent of marten habitat. Burn sizes indicate the proportion of the landscape consumed by a single fire. The color assigned to individual fires (black vs. blue) has no significance.

In the absence of wildfire, the simulated marten population reached steady-state by year 10 (Figure 9), with a consistent mean of ~663 individuals ±15% (min = 562; max = 757). When impacted by wildfire, the mean population size never fully stabilized, but grew to 648 martens by the end of the 50-year time period. Relative to its undisturbed analog, the variability in population size increased dramatically when the martens were subjected to fire. This resulted from reductions in the minimum population size, which when averaged over years 10–50, dropped to just 385 martens. In these worse-case examples, wildfire ended up lowering the simulated marten population by up to 42%.

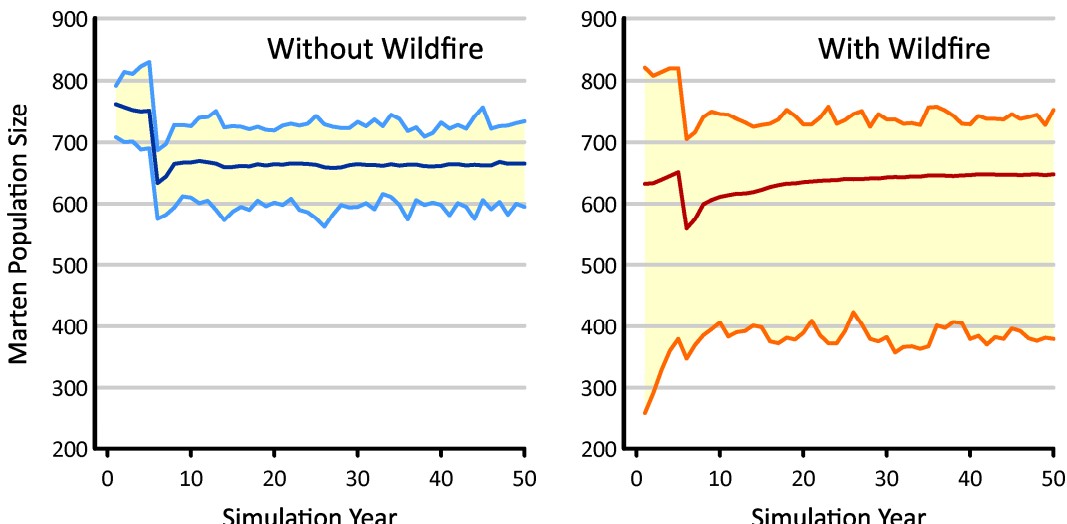

**Figure 9.** Estimates of population size from marten life history simulator, without ((**left**), 100 replicates) and including ((**right**), 1000 replicates) impacts from wildfire. The dark blue and red lines at the plot centers indicate mean population size, while the pairs of light blue and orange lines bracket the range of observed values.

## 4. Discussion

Ecosystems across the globe are increasingly threatened by wildfire, making it more important than ever to incorporate these disturbances into species viability projections and planning. However, doing so has traditionally necessitated using the output from fire simulators as input for plant, animal, or other forecasting models. Linking models in this way is technically challenging, cumbersome, and introduces both computational and quality assurance bottlenecks. Here, we illustrate one avenue for surmounting these difficulties. Our HexFire simulator is straightforward to learn and to parameterize, and because it was coded within HexSim, HexFire can be readily linked to a wide range of connectivity, population viability, gene flow, disease spread, or other ecological response models, thereby simplifying both project design and execution.

HexFire is not as nuanced as the majority of established fire models (e.g., FARSITE, FlamMap, FSim, or ABWiSE), and it cannot be expected to exhibit comparable forecasting accuracy, particularly for single-fire events. However, HexFire is designed to make use of detailed rate of spread (ROS) maps and doing so can add considerable realism to its wildfire projections. The additional incorporation of topography or atmospheric conditions is possible but would require fairly involved model modifications. Nevertheless, this does constitute an important area to focus future work, as local couplings of atmospheric and wildfire-generated conditions can sometimes overwhelm the mechanisms presently captured within HexFire. Advancements such as these would improve HexFire's ability to inform fire suppression activities in real time. While HexSim can easily track multiple continuous-valued state variables, correlated multi-step transition probabilities, and numerous complex interactions, HexFire presently utilizes little of this simulation machinery. Introducing this type of realism would increase HexFire's complexity and make the resulting model harder to learn than the version described here. In contrast, making modifications to HexFire's mechanisms for simulating contagious or ember-driven fire spread could be quite straightforward (see Supplemental Materials).

We made use of three examples to illustrate HexFire's mechanics and emergent behaviors. These demonstrations were intentionally constructed from fabricated data sets and avoided the mention of absolute spatial or temporal scales. We did this so as not to distract from our overall objective: to describe HexFire software design and potential applications. Our first example demonstrated HexFire's sensitivity to model parameters that can modify fire outcomes.

Our second illustration examined an application involving wildfire management. In its present state, HexFire improves upon the capability of most fire models to represent fire suppression strategies. It does so by providing options for representing fuel breaks, mechanisms to adjust their effectiveness, and methods for forecasting the impacts of fuel breaks and back burns on wildfire spread. HexFire allows users to easily modify fuel flammability in space and time using both landscape maps and barrier maps. These model attributes allow users to add nuance to the exploration of fuel break effectiveness, and to distinguish between active and passive fire suppression activities. More generally, they illustrate how HexSim's design simplifies the simulation of complex spatio-temporal dynamics. This capacity is essential for adding realism and relevance to the modern forecasting models used across a range of disciplines. Source code that does these things well is necessarily complicated, and inevitably introduces costs to transparency and transferability. HexFire illustrates how use of the HexSim platform can help insulate users from this complexity.

Our third example demonstrated how HexFire could be an asset to studies focused primarily on ecological endpoints other than fire. This illustration, which involved a simple pairing of HexFire with a realistic lifecycle model, allowed us to draw conclusions about the consequences of wildfire for an already heavily impacted species. Here, HexFire functioned as a disturbance generator that produced an integrated response on the part of a simulated marten population. This loosely coupled system exhibited an increased vulnerability to extinction by the marten, characterized by population reductions of up to 42% in the presence of wildfire.

## 5. Conclusions

We developed HexFire for two principal reasons: first, we wanted to create a general and low-parameter wildfire model that was accessible to researchers who do not specialize in the simulation of fire mechanics; second, we wanted to provide the ecology and conservation communities with a disturbance generator that could be coupled to models that track plant and animal population connectivity, distributions, viability, and other endpoints of concern.

When fire is just one component of a larger system under study, it may not be realistic or even desirable to embrace the complexity of a traditional wildfire simulator. For example, it would not make sense to drive a parsimonious data-poor ecological response model using a highly nuanced and parameter-intensive fire model. We developed HexFire for use in such instances—cases where a simple wildfire disturbance submodel is appropriate for the system being examined. We designed HexFire so that researchers and managers alike could use it to quantify the outcomes generated by multiple possible combinations of fire environments, fuel treatments, fuel breaks, and back burns. In doing so, we have taken advantage of advanced spatio-temporal simulation machinery built into the HexSim platform, which has not previously been introduced to fire ecologists.

**Supplementary Materials:** A 40-min video tutorial about HexFire is available at https://youtu.be/p_ILu2zXDAI (accessed on 10 July 2022). HexSim, HexFire, and all of the content associated with the three examples, may be downloaded from www.hexsim.net (accessed on 10 July 2022). The source code and executables for the three C-language utilities mentioned in the text (Build Patch Maps, Build Wind Map, and Build Average Hexmap) are included with the HexFire workspace.

**Author Contributions:** Conceptualization, N.H.S., S.M.W. and J.A.H.; methodology, N.H.S. and S.M.W.; software, N.H.S. and S.M.W.; validation, N.H.S., S.M.W. and J.A.H.; formal analysis, N.H.S., S.M.W. and J.A.H.; investigation, N.H.S., S.M.W. and J.A.H.; writing—original draft preparation, N.H.S.; writing—review and editing, N.H.S., S.M.W. and J.A.H. All authors have read and agreed to the published version of the manuscript.

**Funding:** This research received no external funding.

**Institutional Review Board Statement:** Not applicable.

**Informed Consent Statement:** Not applicable.

**Data Availability Statement:** Not applicable.

**Acknowledgments:** The information in this document has been funded in part by the U.S. Environmental Protection Agency. It has been subjected to review by the Center for Public Health and Ecological Assessment and approved for publication. Approval does not signify that the contents reflect the views of the Agency, nor does mention of trade names or commercial products constitute endorsement or recommendation for use.

**Conflicts of Interest:** The authors declare no conflict of interest.

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
