# Peer review of "HexFire: A Flexible and Accessible Wildfire Simulator"

_land, doi:10.3390/land11081288_

Round 1

Reviewer 1 Report

The work fits with journal scope very well. The manuscript is clear, relevant for the field and written in well-structured manner.

The topic of the manuscript is current. Applied methodology is original and well defined. Results presented in the manuscript are significant. Those are interpreted appropriatelly, provide an advancement of the current knowledge. The highest standards for results presentation were used. Conclusions are justified and supported by the results.

The study was correctly designed and sound technically good. 

Even the manuscript is well-written, some minor revisions are required:

1. Text editing: Style of References chapter title must be revised.

2. Suplementing the other sources of literature (published in last 5 years)

Final statement: Accept after minor revisions

Author Response

Thank you very much for your thoughtful review of our manuscript.  We apologize for our errors in the formatting of subheadings and citations, and for our not having done a better job of ensuing that our citations were as current as possible.

We have made the following changes to the manuscript in order to correct these shortcomings:

1.  We changed the formatting of all of the manuscript subheadings.  They now correctly follow the policies described in the Instructions for Authors document.

2.  We have added quite a number of more current citations, and have removed some of the older citations.  Now, 55% of our citations are no more than 5 years old.  The total number of citations has increased from 35 to 38.

3.  We have made sure that our self-citation rate is below 15%.  In the revised manuscript, our self-citation rate is presently 13%.

4.  We have used the Mendeley application to completely rebuild our References section.  All of our citations should now correctly follow the Land style guide.

5.  We have also checked the entire manuscript carefully, and made a few small word-choice changes to enhance readability.  We also removed the shading from Table 3, as this was causing the table to reproduce poorly in PDF format.

Reviewer 2 Report

In my opinion, the mathematical model of the occurrence and spread of forest fires proposed by A.M. Grishin. This model was developed and published in the nineties of the last century. And many of the models mentioned in the review use this approach.

Author Response

Thank you very much for your suggestion.  We have included the 1988 publication by A. Grishin in our literature citations.

Reviewer 3 Report

The paper suggests that the model may be suitable for operational purposes. I think this could be clarified, since rates of spread and fire breaks can be overwhelmed by local coupling atmospheric conditions. Having said that I think the paper is suitable for publication in it's present form. 

Author Response

Thank you for these reflections on our treatment of the influence of atmospheric conditions on the model's forecasts.  We have made some changes to our Discussion section to address your suggestions.  Specifically, we have added two new sentences to the second paragraph.  The modified portion of this paragraph now reads as follows (the new text is in italics):

The additional incorporation of topography or atmospheric conditions is possible, but would require fairly involved model modifications. Nevertheless, this does constitute an important area to focus future work, as local couplings of atmospheric and wild-fire-generated conditions can sometimes overwhelm the mechanisms presently captured within HexFire. Advancements such as these would improve HexFire’s ability to inform fire suppression activities in real time.